# Polydatin Ameliorates High Fructose-Induced Podocyte Oxidative Stress via Suppressing HIF-1α/NOX4 Pathway

**DOI:** 10.3390/pharmaceutics14102202

**Published:** 2022-10-16

**Authors:** Hong Ding, Chuanfeng Tang, Wei Wang, Ying Pan, Ruiqing Jiao, Lingdong Kong

**Affiliations:** State Key Laboratory of Pharmaceutical Biotechnology, School of Life Sciences, Nanjing University, Nanjing 210023, China

**Keywords:** polydatin, HIF-1α/NOX4 pathway, podocyte injury, high fructose, oxidative stress

## Abstract

Long-term high fructose intake drives oxidative stress, causing glomerular podocyte injury. Polydatin, isolated from Chinese herbal medicine *Polygonum cuspidatum*, is used as an antioxidant agent that protects kidney function. However, it remains unclear how polydatin prevents oxidative stress-driven podocyte damage. In this study, polydatin attenuated high fructose-induced high expression of HIF-1α, inhibited NOX4-mediated stromal cell-derived factor-1α/C-X-C chemokine receptor type 4 (SDF-1α/CXCR4) axis activation, reduced reactive oxygen species (ROS) production in rat glomeruli and cultured podocytes. As a result, polydatin up-regulated nephrin and podocin, down-regulated transient receptor potential cation channel 6 (TRPC6) in these animal and cell models. Moreover, the data from HIF-1α siRNA transfection showed that high fructose increased NOX4 expression and aggravated SDF-1α/CXCR4 axis activation in an HIF-1α-dependent manner, whereas polydatin down-regulated HIF-1α to inhibit NOX4 and suppressed SDF-1α/CXCR4 axis activation, ameliorating high fructose-induced podocyte oxidative stress and injury. These findings demonstrated that high fructose-driven HIF-1α/NOX4 pathway controlled podocyte oxidative stress damage. Intervention of this disturbance by polydatin could help the development of the therapeutic strategy to combat podocyte damage associated with high fructose diet.

## 1. Introduction

Podocytes form the filtration barrier with glomerular basement membrane and mesangial cells. Nephrin and podocin are the major structural components of glomerular slit diaphragms, which are essential for proper regulation of podocyte structure and function. Podocyte structure injury can cause proteinuria [1]. A high fructose diet is reported to induce reactive oxygen species (ROS) overproduction and weaken antioxidant capacity, possibly reducing glomerular podocyte structural components and exacerbating proteinuria [2,3]. Therefore, antioxidants aimed at maintaining podocyte structure stability may have potential clinical significance.

Polydatin (3,4′,5-trihydroxy-stilbene-3-β-D-glucoside) is a natural precursor of resveratrol, mainly from *Polygonum cuspidatum* Sieb. et Zucc. Polydatin reduces hydrogen peroxide (H_2_O_2_) and superoxide (O_2_^−^) overproduction and prevents kidney dysfunction and fibrosis in diabetic mice [4]. Our previous studies showed that polydatin reduced liver ROS production and improved antioxidant capacity to maintain podocyte structural homeostasis in high fructose-fed rats [5,6]. Polydatin with anti-oxidation may be used to treat multiple organ injury [7,8]. Hypoxia-inducible factor 1α (HIF-1α) is affected by ROS [9,10,11]. HIF-1α deletion prevents glomerulosclerosis in mouse model with podocyte injury [12]. Polydatin is reported to decrease HIF-1α expression in breast cancer cells [13]. However, it remains unclear whether polydatin inhibits HIF-1α to prevent oxidative stress-driven podocyte injury.

C-X-C chemokine receptor type 4 (CXCR4), a G-protein-coupled receptor, is activated through binding to its only known ligand stromal cell-derived factor-1α (SDF-1α). Blocking SDF-1α significantly increases the expression of nephrin and podocin, prevents the onset of albuminuria, and reduces the degree of glomerulosclerosis in mouse model of type 2 diabetes [14]. Thus, SDF-1α/CXCR4 axis dysfunction may participate in the development of kidney injury [15] and rapidly progressive glomerulonephritis [16]. Nicotinamide adenine dinucleotide phosphate oxidase (NOX) is a key enzyme that catalyzes ROS production. NOX4 overexpression activates SDF-1α/CXCR4 axis and aggravates podocyte injury in adriamycin nephropathy of mice [17]. Transient receptor potential cation channel 6 (TRPC6) is functionally connected to podocyte actin cytoskeleton, regulating podocyte skeletal structure and redox homeostasis [18,19]. NOX4 can increase podocyte TRPC6, inducing renal damage in diabetic kidney disease [20]. Podocyte-specific NOX4 knockout also reduces albuminuria, and improves glomerular sclerosis, mesangial expansion, and glomerular basement membrane thickness in streptozotocin-induced diabetes of mice [21]. High fructose feeding increases NOX4 expression in renal cortex of rat and mouse livers [22,23]. NOX4 is reported as the target gene of HIF-1α in mouse smooth muscle cells [24], but it is unknown in glomerular podocytes.

Here, we focused on the relationship of HIF-1α and NOX4 in glomerular podocytes under oxidative stress driven by high fructose. We found that high fructose increased NOX4 expression to aggravate SDF-1α/CXCR4 axis activation in a HIF-1α-dependent manner, which was sufficient to induce oxidative stress and glomerular podocyte structure damage. Considering the important role of antioxidants in ameliorating glomerular podocyte injury [25], we found that polydatin down-regulated HIF-1α to inhibit NOX4 and then suppressed SDF-1α/CXCR4 axis, preventing high fructose-induced podocyte oxidative stress injury. Additionally, we demonstrated that the intervention of the high fructose-driven HIF-1α/NOX4 pathway could provide a promising strategy for treating glomerular podocyte damage in clinic.

## 2. Materials and Methods

### 2.1. Animals and Treatments

Male Sprague-Dawley rats (6–7 weeks old, 180–220 g of body weight) were purchased from the Experimental Animal Centre of Zhejiang Province (Hangzhou, China). The experimental animals were housed in the animal room of the College of Life Sciences of Nanjing University with a room temperature of 22 ± 2 °C, a humidity of 55 ± 5%, and a 12 h light/dark cycle (light from 9:00 a.m. to 9:00 p.m.). All rats were given standard chow and water (ad libitum) 7 days for acclimatization before the experiment, and then randomly divided into 6 subgroups (*n* = 8): normal-control, fructose-vehicle, fructose with polydatin (15721, Sigma-Aldrich, Shanghai, China) (7.5, 15, and 30 mg/kg) as well as fructose with allopurinol (A8003, Sigma-Aldrich, Shanghai, China) (positive drug, 5 mg/kg). Polydatin or allopurinol was mixed with normal saline by ultrasonic waves, and then suspended or dissolved. The rats of normal control group had standard chow and water for 13 weeks. The remaining five groups of rats were given 10% fructose (Shandong Xiwang Sugar Industry Co., Ltd., Binzhou, China) (wt/vol) water for 6 weeks, followed by the treatment with saline, polydatin, or allopurinol by intragastric gavage for the next 7 weeks, respectively, according to our previous report [5]. Allopurinol, as an effective xanthine oxidase (XO) inhibitor, reduces serum uric acid levels and slows down the progression of renal disease in patients with chronic kidney disease [26,27]. Our previous study showed that allopurinol protected against high fructose-induced apoptosis in rats and cultured podocytes [28]. Therefore, allopurinol was used as a positive drug to evaluate the effect of polydatin on high fructose-induced podocyte injury in this study. All drugs were administered once daily at 2:30–3:30 p.m. Doses of polydatin [5,29,30] and allopurinol [28,31,32] were selected based on our previous studies and other reports. Body weight was recorded weekly throughout the experiment. At week 12, each rat was placed in a metabolic cage to collect 24-h urine, respectively. Urine samples were centrifuged at 3000× *g* for 10 min at 4 °C to remove particulate contaminants and stored at −80 °C. Animal welfare and experimental procedures were carried out following the criteria outlined in the Guide for the Care and Use of Laboratory Animals enacted by the National Academy of Sciences and published by the National Institutes of Health (NIH publication 86-23 revised 1985) and the related ethnical regulations of Nanjing University [SYXK (SU) 2009-0017]. All efforts were made to minimize animals’ suffering and to reduce the number of animals used.

### 2.2. Blood and Tissue Sample Collection

At the end of the animal experiment, rats were anesthetized intraperitoneally using sodium pentobarbital (50 mg/kg). Blood samples were collected from rat carotid arteries. The blood samples were centrifuged at 3000× *g* for 10 min at 4 °C to get the serum and then stored at −80 °C for biochemical assays. Rat kidney cortex tissues were cut quickly into equal pieces on ice for glomeruli isolation, Western blot, immunofluorescence staining, quantitative real-time PCR (qRT-PCR) analysis, etc. 

Glomeruli were isolated by conventional sieving as described previously [28]. Briefly, kidney cortex tissues were cut into small pieces and pressed through 250 μm and 150 μm stainless steel mesh and then thoroughly washed with 0.01 M phosphate-buffered saline (PBS, pH 7.4) (C10010500BT, Gibico, Grand Island, NY, USA) on 75 μm stainless meshes. The rat kidney cortex tissues were fixed with 4% paraformaldehyde. The kidney cortex tissues and glomeruli samples were stored at −80 °C for further analysis.

### 2.3. Immunofluorescence Assay

Immunofluorescence staining was performed with the specific podocyte markers nephrin, podocin, and synaptopodin for localization. The detailed steps were performed as described previously [33]. Briefly, rat kidney cortex tissues were fixed with 4% paraformaldehyde, embedded in paraffin, and sectioned transversely. These cortex tissues were blocked for 1 h with immunostaining blocking buffer and then incubated with primary antibodies overnight at 4 °C. The types and concentrations of the primary antibodies were as follows: anti-synaptopodin (sc-515842, 1:100, Santa Cruz, Dallas, TX, USA), anti-nephrin (ab58968, 1:100, Abcam, Cambridge, MA, USA) and anti-podocin (ab50399, 1:100, Abcam, Cambridge, MA, USA). After washing with 0.01 M PBS, the sections were incubated with the secondary antibodies including Alexa Fluor^®^ 555 goat anti-mouse IgG (A21422, 1:500, Life Technologies, Gaithersburg, MD, USA), Alexa Fluor^®^ 488 goat anti-rabbit IgG (A11008, 1:500, Life Technologies, Gaithersburg, MD, USA) for 1 h at room temperature, and then stained with DAPI (4’,6-diamidino-2-phenylindole, ab228549, 1:2000, Abcam, Cambridge, MA, USA), at room temperature for 10 min. Finally, the sections were examined by a confocal laser scanning microscope (Lei TCS SP8-MaiTai MP; Leica, Wetzlar, Germany).

### 2.4. Cell Culture and Treatment

The heat-sensitive human podocyte cell line was presented by Dr. Zhi-Hong Liu (Research Institute of Nephrology, Nanjing General Hospital of Nanjing Military Command, Nanjing, China), provided by M. Saleem (University of Bristol, Bristol, United Kingdom). The podocytes were cultured in RPMI-1640 medium (C11875500BT, Gibico, Grand Island, NY, USA) supplemented with 10% fetal bovine serum (FBS) (Wisent, St-Bruno, QC, Canada) and recombinant interferon-γ (IFN-γ) (CAA31639, R&D Systems, Minneapolis, MN, USA) at the permissive temperature (33 °C). To induce differentiation, podocytes were plated in 6-well plastic culture plates at the density of 5 × 10^4^ cells/mL, and grown under nonpermissive conditions at 37 °C for 5–7 days in the absence of IFN-γ. The medium was changed as appropriate. The cell culture method was based on our previous report [28].

Differentiated podocytes were cultured in RPMI 1640 medium (containing 10% FBS) supplemented with or without 5 mM fructose for different times in the presence or absence of polydatin (20, 40, and 80 μM) or allopurinol (100 μM) to detect HIF-1α (12 h), NOX4 (24 h), SDF-1α (72 h), CXCR4 (72 h), TRPC6 (72 h), nephrin (72 h), and podocin (72 h) protein levels by Western blot, respectively. The selected concentrations of fructose, polydatin, and allopurinol were in accordance with our previous studies [5,28].

*HIF-1α* siRNA and the negative control were synthesized by Biotend (Shanghai, China). The sequences were listed in Table 1. Transfection of *HIF-1α* siRNA (50 nM) or negative control in differentiated podocytes was performed using Lipofectamine 2000 (11668-019, Invitrogen, Carlsbad, CA, USA), according to the manufacturer’s instructions. After transfection, the efficiency of transfection was evaluated by measuring mRNA levels at 24 h by qRT-PCR, and protein levels at 48 h by Western blot. After the transfection of *HIF-1α* siRNA for 6 h, transfected cells were incubated in RPMI 1640 medium supplemented with 10% FBS and 5 mM fructose in the presence or absence of 80 μM polydatin or 100 μM allopurinol to detect HIF-1α, NOX4, SDF-1α, CXCR4, nephrin, podocin, and TRPC6 protein levels in 48 h by Western blot, respectively.

Differentiated podocytes were cultured in RPMI 1640 medium (containing 10% FBS) supplemented with or without 5 mM fructose in the presence or absence of polydatin (80 μM) for 12 h, then incubated with 10 μM cycloheximide (CHX, FS0076, Fushen Biotechnology, Shanghai, China), a protein synthesis inhibitor for 2, 4, or 8 h to detect HIF-1α by Western blot, respectively. The selected concentration of CHX was in accordance with a previous study [34].

In all experiments, cell lysates were collected, and total cellular protein or RNA was extracted, respectively. These samples were store d at −80 °C until assays.

### 2.5. Biochemical Analysis

Intracellular ROS and O_2_^−^ levels in podocytes were detected using the fluorescence probe DCFH_2_-DA (287810, Sigma-Aldrich, Shanghai, China) and fluorescence probe DHE (D7008, Sigma-Aldrich, Shanghai, China), respectively. In short, podocytes were plated in a 6-well (5 × 10^4^ cells/mL) or 96-well clear-bottom microplate (1 × 10^4^ cells/mL). After the differentiation and treatment above, podocytes were washed twice with 0.01 M PBS (pH 7.4) and then incubated in RPMI 1640 medium containing 10 μM DCFH_2_-DA or 10 μM DHE without FBS for 30 min at 37 °C. Cells were analyzed by Attune™ NxT flow cytometry (Thermo Fisher Scientific, Waltham, MA, USA). A total of 10,000 events were collected for each group, and the relative ROS and O_2_^−^ levels were analyzed using FlowJo software v10.8 (Ashland, KY, USA). Fluorescence intensity of DCFH_2_-DA was measured at λ_ex_ = 488 nm and λ_ex_ = 525 nm, and DHE was measured at λ_ex_ = 535 nm and λ_ex_ = 620 nm with a microplate reader. Then, these cell samples were measured for cell viability by using the CCK-8 kit (C0039, Jiancheng Bioengineering Institute, Nanjing, China). The samples were cultured with 10 μL of CCK-8 solution at 37 °C for another 1 h. Finally, the absorbance was read at λ_ex_ = 450 nm with a plate reader. Cell activity was calculated and normalized to the data of ROS or O_2_^−^ levels in podocytes.

NADP^+^/NADPH (S0179) in podocytes were measured by commercially available biochemical kits from Beyotime Biotechnology (Shanghai, China), Superoxide dismutase activity (SOD) (A001-3-2, Jiancheng Bioengineering Institute, Nanjing, China), and catalase (CAT) activity (A007-1-1, Jiancheng Bioengineering Institute, Nanjing, China) in podocytes were measured by commercially available biochemical kits according to the manufacturer’s instructions, respectively.

Malondialdehyde (MDA) (A003-4-1), SOD (A001-3-2), and CAT (A007-1-1) activity in rat glomeruli were measured by commercially available kits (Jiancheng Bioengineering Institute, Nanjing, China), using a bicinchoninic acid (BCA) kit (23225, Thermo Fisher Scientific, Waltham, MA, USA) for protein quantification to correct the detection results.

Uric acid (C012-2-1), creatinine (C011-1-1), urea nitrogen (C013-2-1) levels in serum and urine protein (C035-2-1) concentration were measured by commercially available biochemical kits (Jiancheng Bioengineering Institute, Nanjing, China), respectively.

Kidney tissue in situ O_2_^−^ production was detected by fluorescence probe DHE labeling. Briefly, 10 μm-thick frozen sections were incubated with 10 μM DHE probe for 30 min, 37 °C in a humidified chamber protected from light, rinsed in PBS, and analyzed by using confocal laser scanning microscope. Fluorescent images were quantified by counting the percentage of positive nuclei from total nuclei per glomeruli cross-section by Image J (National Institutes of Health, Bethesda, MD, USA).

### 2.6. RNA Isolation and qRT-PCR Analysis

Total RNA was isolated from differentiated podocytes or rat glomeruli using Trizol reagent (R401-01-AA, Vazyme Biotech Co., Ltd., Nanjing, China) according to the manufacturer’s recommendations. All the primers were provided by GENEray Biotechnology (Shanghai, China), and listed in Table 1.

The process of reverse transcription from RNA to single-stranded cDNA requires RNA 2 μL (0.5 μg), 5 × HiScript II Select qRT SuperMix 2 μL (R222-01, Vazyme Biotech Co., Ltd., Nanjing, China), and ddH_2_O 6 μL. The reaction mixture was incubated for 15 min at 50 °C and 5 min at 85 °C and then stored at 4 °C. The complementary DNA amplification process requires cDNA 1 μL, iTaqTm Universal SYBR Green Supermix 5 μL (Q321-01, Vazyme Biotech Co., Ltd., Nanjing, China), diethylpyrocarbonate (DEPC)-H_2_O (BL510B, Biosharp life sciences, Hefei, China) 3.5 μL, primers 0.5 μL (forward and reverse primers 10 μM, respectively). The reaction mixture was amplified in 96-well optical reaction plates. qRT-PCR was carried out according to the following reaction procedure: 40 cycles of 94 °C for 30 s, 95 °C for 5 s, and 60 °C for 30 s on a real-time PCR detection system (Bio-Rad CFX96 Real-Time PCR Detection System). The specificity of the amplification was confirmed using a melting curve analysis. Data were collected and recorded by CFX Manager Software (Bio-Rad, Hercules, CA, USA), and expressed as a function of threshold cycle (Ct). The samples for qRT-PCR analysis were evaluated using a single predominant peak as quality control. Relative expressions of target genes were determined by the Ct (2^−ΔΔCt^) method and normalized to β-actin, respectively. All reactions were performed in triplicates for each sample.

### 2.7. Western Blot Analysis

Rat glomeruli was homogenized in ice-cold RIPA buffer (P0013C, Beyotime Institute of Biotechnology, Shanghai, China) containing 0.1 mM phenylmethanesulfonyl fluoride (PMSF) (10837091001, Sigma-Aldrich, Shanghai, China), and then centrifuged at 12,000× *g* for 15 min at 4 °C. The protein samples of cultured podocytes were prepared in cell lysis buffer (P0013, Beyotime Institute of Biotechnology, Shanghai, China), containing 0.1 mM PMSF, and then centrifuged at 12,000× *g* for 15 min at 4 °C. BCA kit was used to correct the protein sample concentration. Equivalent amounts of protein from each sample were separated by 10% SDS-PAGE and transferred onto polyvinylidene fluoride membranes (IPVH100010, Millipore Ltd., Boston, MA, USA). Then, the membranes were incubated in Tris-HCl buffer solution containing 0.1% Tween-20 and 5% fat-free milk for 1 h to block nonspecific protein-binding sites at room temperature, probed overnight at 4 °C with primary antibodies, and then incubated with HRP-conjugated secondary goat anti-mouse (SA00001-1, Proteintech Group, Inc, Chicago, IL, USA) or goat anti-rabbit (SA00001-2, Proteintech Group, Inc, Chicago, IL, USA) antibodies. The primary antibodies used in this study included anti-nephrin (ab58968, 1:1000), anti-podocin (ab50399, 1:1000), anti-HIF-1α (ab2185, 1:1000), anti-NOX4 (ab133303, 1:1000), anti-SDF-1α (ab18919, 1:500), anti-CXCR4 (ab124824, 1:1000), and anti-TRPC6 (ab62461, 1:1000) and were purchased from Abcam (Cambridge, MA, USA). The primary antibody of anti-glyceraldehyde-3-phosphate dehydrogenase (GAPDH) (sc-25778, 1:2000) was purchased from Santa Cruz (Dallas, TX, USA). The primary antibody of anti-β-actin (ABM-001-100, 1:5000) was purchased from Zoonbio Biotechnology (Nanjing, China).

The immunoreactive protein was detected by a high enhanced chemiluminescence Western blot peroxide buffer (180–5001 W, Tanon Science & Technology Co., Ltd., Shanghai, China), and then, the signals were visualized with an enhanced chemiluminescence system (Tanon Science & Technology Co., Ltd., Shanghai, China). The immunoreactive bands were quantified via densitometry using Image J, standardized to β-actin or GAPDH, and expressed as fold changes relative to the control value.

### 2.8. Statistical Analysis

Data were expressed in Mean ± standard error of the mean (S.E.M). Unpaired t test was used for comparisons between two groups. Statistical analysis of multiple groups was performed by ANOVA with Dunnett’s multiple comparisons test, using GraphPad Prism 8.0 software (San Diego, CA, USA). *p* < 0.05 was considered statistically significant.

## 3. Results

### 3.1. Polydatin Inhibits the Overexpression of HIF-1α in High Fructose-Cultured Podocytes

Previous study has shown that polydatin reduced high fructose-induced ROS and O_2_^−^ levels in cultured podocytes and rat glomeruli [6]. In addition, there is a direct regulatory relationship and mechanism between ROS and HIF-1α [9]. To explore whether there is a correlation between HIF-1α and the antioxidant effect of polydatin on podocyte damage, we detected changes in HIF-1α at both transcriptional and translational levels. HIF-1α mRNA and protein levels were increased obviously in rat glomeruli and cultured podocytes under high fructose stimulation, which were reversed by polydatin (Figure 1A–D). To determine the effect of high fructose on the degradation of HIF-1α, we examined the protein half-life of HIF-1α incubated with fructose, and found that high fructose increased HIF-1α expression, but did not affect its degradation in podocytes. Moreover, polydatin had no effect on HIF-1α degradation (Figure 1E). These results indicated that HIF-1α may be the target protein of polydatin, possibly contributing to its suppression of high fructose-induced oxidative stress.

### 3.2. Polydatin Inhibits HIF-1α to Ameliorate Oxidative Stress in High Fructose-Cultured Podocytes

To verify the certain inhibitory effect of polydatin on podocyte oxidative stress, redox homeostasis analysis was further carried out. As expected [4,5,6], polydatin significantly reduced ROS (Figure 2A,B), O_2_^−^ (Figure 2C,D), and NADP^+^/NADPH (Figure 2E), while increasing SOD (Figure 2F) and CAT (Figure 2G) activity in high fructose-cultured podocytes, further demonstrating that polydatin inhibited high fructose-induced ROS overproduction and enhanced the antioxidant capacity. Next, we investigated the key influence of HIF-1α on podocyte oxidative stress. We transiently transfected podocytes with *HIF-1α* siRNA and then incubated them with or without fructose, respectively. qRT-PCR and Western blot analysis were used to confirm high transfection efficiency (Figure 2H,I). High fructose did not increase the expression of HIF-1α in podocytes with the transfection of *HIF-1α* siRNA, while polydatin further down-regulated HIF-1α (Figure 2J). Inhibition of HIF-1α expression by *HIF-1α* siRNA suppressed the accumulation of ROS (Figure 2K), and polydatin potentiated this suppression in podocytes stimulated with high fructose. Therefore, it reminded us that HIF-1α might directly contribute to high fructose-induced ROS production. Polydatin may target HIF-1α to down-regulate its expression to maintain redox balance in high fructose-exposed podocytes.

### 3.3. Polydatin Suppresses HIF-1α/NOX4 Pathway Activation in High Fructose-Induced Oxidative Stress Damage in Podocytes

NOX4 overexpression promotes podocyte injury and exacerbates albuminuria in animal models of diabetes [20,21,35]. Previous studies have shown complex regulatory relationships between NOX4 and HIF-1α [36,37], but the regulatory mechanism of NOX4 in high fructose-induced podocyte damage is unknown. Therefore, we examined the expression of NOX4 and downstream pathway proteins, including SDF-1α and CXCR4 in the cell model. The protein levels of NOX4, SDF-1α, and CXCR4 were significantly increased in high fructose-exposed podocytes (Figure 3A–C). To investigate whether HIF-1α contributed to NOX4-mediated redox status imbalance in podocytes, we tested the NOX4 pathway in high fructose-exposed podocytes with transfection of *HIF-1α* siRNA. *HIF-1α* siRNA abolished high fructose-induced up-regulation of NOX4 (Figure 3D), SDF-1α (Figure 3E), and CXCR4 (Figure 3F) at protein levels in podocytes. In addition, the protein levels of structural proteins nephrin and podocin were decreased, while of TRPC6 were increased in this cell model (Figure 4A–C). Knockdown of HIF-1α restored the expression levels of these structural proteins (Figure 4D–F) in high fructose-exposed podocytes. These results indicated that HIF-1α accumulation-induced NOX4 over-expression may mediate SDF-1α/CXCR4 axis activation, resulting in high fructose-induced podocyte injury.

Polydatin markedly decreased NOX4 expression and down-regulated SDF-1α and CXCR4 in cultured podocytes stimulated with high fructose (Figure 3A–C). The abnormal expression of nephrin, podocin, and TRPC6 induced by high fructose was also attenuated by polydatin and allopurinol (Figure 4A–C). Moreover, polydatin did not significantly reduce NOX4, SDF-1α, and CXCR4 protein levels in high fructose-exposed podocytes with the transfection of *HIF-1α* siRNA (Figure 3D–F). Meanwhile, the expression of nephrin, podocin, and TRPC6 was not significantly altered (Figure 4D–F). These results indicated that polydatin may restore HIF-1α/NOX4 mediated-redox status imbalance to reduce high fructose-induced damage of glomerular podocytes.

### 3.4. Polydatin Prevents Glomeruli Oxidative Stress in High Fructose-Fed Rats with Kidney Dysfunction

Here we utilized high fructose-fed rat model to examine the effect of polydatin on oxidative stress via suppressing HIF-1α/NOX4 pathway. As expected [6], polydatin reduced the accumulation of O_2_^−^ and MDA, and increased the activity of SOD and CAT in high fructose-fed rat glomeruli (Figure 5A–E). Polydatin down-regulated the protein levels of NOX4 (Figure 5F), SDF-1α (Figure 5G), and CXCR4 (Figure 5H) in glomeruli of high fructose-fed rats.

Furthermore, polydatin dramatically up-regulated nephrin and podocin, and down-regulated TRPC6 in high fructose-fed rat glomeruli (Figure 6A–D). Consistently, they significantly improved kidney function evidenced by lowering serum uric acid, creatinine, urea nitrogen levels (Table 2), and urinary protein levels (Figure 6E), while increasing urine creatinine (Table 2) in this animal model. These data showed that polydatin effectively maintained the structural stability of glomerular podocytes and renal function through antioxidant effects in high fructose-fed rats.

## 4. Discussion

Evidence is mounting that high fructose induces redox status imbalance and related kidney disorders [38]. Podocyte injury-caused proteinuria is the main feature of chronic kidney diseases [39]. Polydatin offers renal protection in acute kidney injury and diabetic nephropathy by inhibiting oxidative stress [40,41]. In this study, we firstly found the pathological mechanism by which ROS-driven HIF-1α controlled NOX4 in high fructose-induced podocyte damage. Furthermore, we demonstrated that the inhibition of the HIF-1α/NOX4 pathway might be the crucial link for polydatin to exert its protective effect on podocyte oxidative stress damage (Figure 7).

The regulatory role of HIF-1α on oxidative stress has attracted wide attention, existing controversial findings in many studies [42,43,44]. On the one hand, direct prolyl hydroxylase inhibition and transcriptional and translational regulation of HIF-1α mediated by ROS can prompt HIF-1α stabilization under a normoxic condition [9,45,46]. On the other hand, HIF-1α overexpression is reported to alleviate ROS and apoptosis in benzene-induced hematotoxicity of K562 cells [47]. Prolyl hydroxylase inhibitor counteracts renal metabolic alteration, the response associated with the improvement of oxidative stress in a mouse model of early diabetic nephropathy [48]. On the contrary, glucose enhances ROS production and HIF-1α expression, leading to podocyte injury and fibrosis in diabetes mice [49]. Thus, the positive or negative correlation between ROS and HIF-1α in high fructose-induced podocyte oxidative stress damage is worth exploring. In this study, we found that high fructose-induced oxidative stress was accompanied by high expression of HIF-1α in rat glomeruli and cultured podocytes, which were attenuated by polydatin. Here, it was worth considering whether the high protein level of HIF-1α was the result of the up-regulation of expression or the decrease of degradation in high fructose-cultured podocytes. We found that high fructose induced HIF-1α high expression without affecting its degradation. Taken together, high fructose might stably upregulate HIF-1α through ROS overproduction in podocytes. Polydatin inhibited the high expression of HIF-1α without affecting its degradation in podocytes stimulated by high fructose. It reminded us that HIF-1α expression might be regulated by polydatin at the transcriptional level by some transcription factors, such as nuclear factor kappa-B [50,51] and signal transducer, activator of transcription 3 [52,53]. Furthermore, *HIF-1α* siRNA partially eliminated the accumulation of ROS, indicating a positive regulatory loop that ROS may induce HIF-1α high-expression, which in turn stimulate further overproduction of ROS in podocytes cultured with high fructose. HIF-1α expression was not eliminated in podocytes transfected with *HIF-1α* siRNA. Polydatin further reduced the expression of HIF-1α, and restored ROS overproduction to normal level based on *HIF-1α* siRNA transfection in high fructose-cultured podocytes. Thus, polydatin may target HIF-1α to ameliorate high fructose-induced oxidative stress in podocytes.

The function of HIF-1α in the development of chronic kidney disease is disputed. HIF-1α deficiency may accelerate renal injury in the early stage of streptozotocin-induced diabetes in mice [54]. HIF-1α stabilization alleviated the metabolic reprogramming associated with renal dysfunction in an animal model of diabetic kidney disease [48]. Nevertheless, knockout of HIF-1α was protective against glomerulosclerosis and glomerular type-I collagen accumulation in a mouse podocyte ablation model [12]. Knockout of the Von Hippel–Lindau gene in podocytes is reported to cause the stabilization of HIF-1α, necrotizing crescentic glomerulonephritis in mice [55]. These observations remind us that HIF-1α may play various roles in different development stages of kidney injury. In this study, we observed high HIF-1α expression significantly in high fructose-exposed podocytes and fructose-fed rat glomeruli, being consistent with the down-regulation of nephrin and podocin, as well as the up-regulation of TRPC6. The results from *HIF-1α* siRNA supported that HIF-1α low-expression up-regulated nephrin and podocin and down-regulated TRPC6 to prevent high fructose-induced podocyte damage, indicating the harmful effect of high HIF-1α expression in podocyte function. Moreover, polydatin reversed high fructose-induced down-regulation of nephrin and podocin, and up-regulation of TRPC6, offering protection against podocyte damage. TRPC6 is a nonselective Ca^2+^ channel protein. Its over-activation increases Ca^2+^ influx, resulting in excessive contraction of the podocyte cytoskeleton, which in turn changes the filtration gap between the foot processes and glomeruli filtration function [56]. The slit diaphragm complex formed by TRPC6, nephrin, podocin, and CD2AP directly or indirectly interacts with α-actinin-4 to maintain the structural and functional integrity of slit diaphragm in podocytes [57,58], indicating that abnormal expression of TRPC6 could cause podocyte structural change, leading to impaired filtration barrier and massive proteinuria. High expression of TRPC6 is detected in kidney diseases with proteinuria as the main clinical manifestation (such as glomerular minimal change disease, membranous nephropathy, and focal segmental glomerulosclerosis) [59], as well as in some secondary glomerular diseases (such as diabetic nephropathy) [60]. Some drugs (sildenafil and tacrolimus) and the herb compound astragaloside IV protect kidney function by inhibiting TRPC6 and nuclear factor of activated T cell (a transcription factor targeting TRPC6) [61,62,63]. Polydatin inhibited TRPC6 in high fructose-fed rats, indicating that polydatin may hold promise for treating podocyte structure damage. These results demonstrated that the stable and high expression of HIF-1α might accelerate high fructose-induced podocyte injury, while polydatin abolishes the harmful effects by down-regulating HIF-1α.

In addition to these vital structure proteins of podocytes described above, SDF-1α as a critical element of a podocyte-progenitor feedback axis may inhibit the regeneration of lost podocytes during glomerular injury [64]. The up-regulation of CXCR4 is also detected in kidney biopsy tissues of patients with proteinuria and co-localized with oxidized protein products in clinic [17]. In this study, SDF-1α and CXCR4 had been shown high expression in vivo and in vitro. SDF-1α/CXCR4 axis is reported to be the downstream pathway of NOX4 [17], which catalyzes the formation of H_2_O_2_ in podocytes [65,66]. It is worth noting that there is a relationship between NOX4 and HIF-1α, but which one is upstream regulator is still an open question. HIF-1α may bind to the NOX4 promoter at the hypoxia-responsive element to enhance promoter activity, affecting cell proliferation in the hypoxic mouse pulmonary artery smooth muscle cells [24]. However, inhibition of NOX4 decreases insulin-stimulated HIF-1α expression in human microvascular endothelial cells [67]. The up-regulation of NOX4 promotes tumor angiogenesis by stabilizing HIF-1α in mice [68]. In this study, we found that high fructose-induced high NOX4 expression was significantly inhibited in podocytes transfected with *HIF-1α* siRNA, indicating that NOX4 was the target gene of HIF-1α in podocytes under high fructose condition. Polydatin also showed suppressive effect on NOX4 in high fructose-induced podocyte damage through the inhibition of HIF-1α. Moreover, after the transfection of *HIF-1α* siRNA, the up-regulation of SDF-1α and CXCR4 was eliminated in podocytes cultured with high fructose, indicating that high fructose may induce HIF-1α/NOX4 pathway to activate SDF-1α/CXCR4 axis, possibly contributing to podocyte oxidative damage. The results from this study showed that polydatin may inhibit the HIF-1α/NOX4 pathway to block SDF-1α/CXCR4 axis activation, resulting in protection against high fructose-induced podocyte injury.

Polydatin has various pharmacological activities such as anti-inflammation [51], anti-oxidation [69,70,71], anti-fibrosis [72], and regulation of glucose and lipid metabolism [73,74], protecting multiple tissues and organs from injury. Polydatin can be absorbed well due to its glycoside and enter cells by passive diffusion and active transport [75]. In this study, we explored the interaction between HIF-1α and polydatin in glomerular podocytes, showed that the podocyte protection offered by polydatin is due to the inhibition of the HIF-1α/NOX4 pathway that reduces oxidative stress. This suggested to us the possibility that polydatin may act on HIF-1α in glomerular podocytes.

Allopurinol is used to lower serum uric acid for the treatment of patients with gout and chronic kidney disease [76]. Most clinical trials show that allopurinol significantly increases the glomerular filtration rate and enhances kidney function in young and old chronic kidney disease patients [27,77]. These similar beneficial effects also appear in diabetic patients [78]. Recently, two trial reports showed inconsistent results. The treatment with allopurinol failed to slow the glomerular filtration rate decline in patients with chronic kidney disease or diabetes [79,80]. Altogether, the role of allopurinol in the treatment of kidney diseases is still not well established because of the influence of the patient’s age, basic glomerular filtration rate, uric acid level, medication time, and disease stage. Of note, allopurinol is reported to stabilize cardiac HIF-1α and heme oxygenase 1 protein expression, conferring synergistic cardioprotection against myocardial ischemia/reperfusion injury in diabetic rats [81]. It also reduces HIF-1α expression in human foreskin fibroblasts and human umbilical vein endothelial cells under hypoxia and normoxia [82]. In this study, high fructose-induced high expression of HIF-1α was abolished by allopurinol in podocytes and rat glomeruli. NOX and XO mediate two parallel oxidation pathways as sources of ROS. There are few studies on the regulatory relationship between XO inhibitor and NOX. In this study, we found that allopurinol down-regulated NOX4 in vivo and in vitro, indicating that allopurinol may suppress high fructose-induced high expression of NOX4 through inhibiting HIF-1α. High uric acid induces SDF-1α in human kidney-2 cells [83]. Here, this study showed that allopurinol restrained the activation of SDF-1α/CXCR4 axis induced by high fructose in podocytes and rat glomeruli. As expected [28], allopurinol increased nephrin and podocin, being consistent with the down-regulation of TRPC6 to stabilize podocyte structure in these animal and cell models. These results suggest that allopurinol may not only reduce NOX4 expression by suppressing HIF-1α but also have a new role in inhibition of SDF-1α/CXCR4 axis activation and down-regulation of TRPC6 to prevent podocyte structure damage under high fructose stimulation.

In this study, we demonstrated that high-fructose-increased HIF-1α expression exacerbated redox status imbalance and podocyte damage by promoting NOX4 to activate SDF-1α/CXCR4 axis. Polydatin restrained HIF-1α expression to inhibit NOX4 and suppressed SDF-1α/CXCR4 axis activation, resulting in the attenuation of high fructose-induced oxidative stress damage of glomerular podocytes.

## 5. Conclusions

This study provides evidence that the specific interaction between HIF-1α and NOX4 determined redox balance in podocytes under high fructose condition. The inhibition of SDF-1α/CXCR4 axis mediated by HIF-1α/NOX4 pathway suppression might attenuate high fructose-induced podocyte structural injury. Polydatin alleviated oxidative stress-related podocyte damage by restraining the HIF-1α/NOX4 pathway, showing its anti-oxidative effects in a new way. Overall, these findings elucidated that the inhibition of the HIF-1α/NOX4 pathway by polydatin may be a novel treatment strategy for podocyte injury, and further suggest the potential of the antioxidant agent polydatin to be effective in the prevention and treatment of kidney injury.

## Figures and Tables

**Figure 1 pharmaceutics-14-02202-f001:**
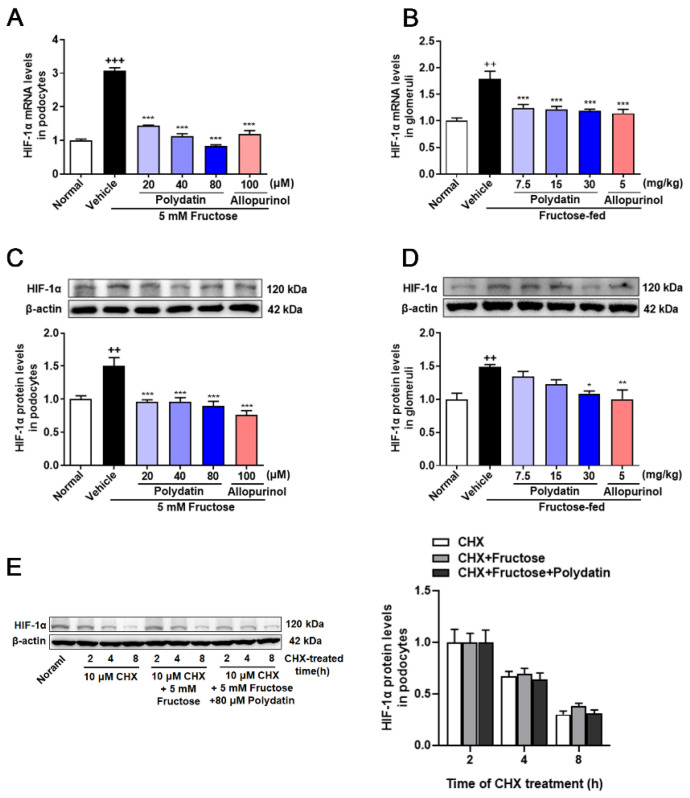
Polydatin inhibits the overexpression of HIF-1α in high fructose-cultured podocytes. (**A**,**B**) Relative mRNA levels of HIF-1α were detected in podocytes (*n* = 4–6) and rat glomeruli (*n* = 5–8) by qRT-PCR. (**C**,**D**) Relative protein levels of HIF-1α were detected in podocytes (*n* = 6–8) and rat glomeruli (*n* = 4–5) by Western blot and quantified by Image J. (**E**) Relative protein levels of HIF-1α were detected in podocytes stimulated with or without fructose (5 mM), polydatin (80 μM), and allopurinol (100 μM) for 12 h and then incubated 10 μM cycloheximide for 2, 4, or 8 h (*n* = 3) by Western blot and quantified by Image J. Protein levels were normalized to the fold of 2 h. Data are expressed as the mean ± S.E.M. ^++^
*p* < 0.01, ^+++^
*p* < 0.001 compared with normal control group, * *p* < 0.05, ** *p* < 0.01, *** *p* < 0.001 compared with fructose vehicle group.

**Figure 2 pharmaceutics-14-02202-f002:**
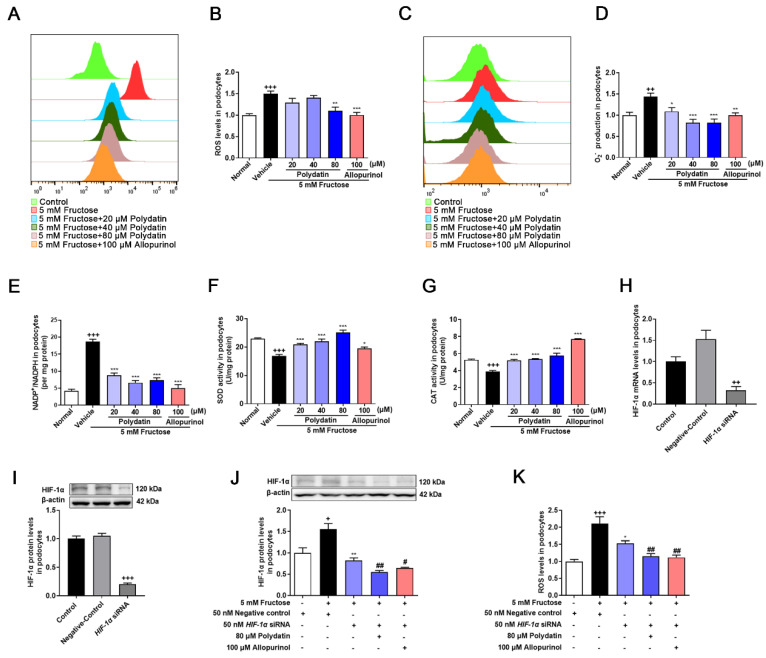
Polydatin inhibits HIF-1α to ameliorate oxidative stress in high fructose-cultured podocytes. Podocytes were cultured with or without fructose (5 mM) in the presence or absence of polydatin (20, 40, and 80 μM) or allopurinol (100 μM). (**A**,**B**) ROS levels (24 h) were analyzed by labeling fluorogenic probe DCFH_2_-DA, evaluated by flow cytometry (*n* = 3) and microplate reader (*n* = 5) in podocytes. (**C**,**D**) O_2_^−^ levels (24 h) were measured by fluorogenic probe DHE, evaluated by flow cytometry (*n* = 3) and microplate reader (*n* = 7–10) in podocytes. (**E**–**G)** Intracellular NADP^+^ and NADPH levels (48 h) (*n* = 4) (**E**), SOD activity (48 h) (*n* = 8–12) (**F**), and CAT activity (48 h) (*n* = 6–9) (**G**) in podocytes were measured, respectively. (**H**,**I**) Relative mRNA and protein levels of HIF-1α in podocytes transfected with *HIF-1α* siRNA were detected by qRT-PCR (*n* = 2–3) and Western blot (*n* = 7–8). (**J**) Relative protein levels of HIF-1α were detected in podocytes transfected with *HIF-1α* siRNA and then incubated with or without fructose (5 mM), polydatin (80 μM), and allopurinol (100 μM) by Western blot (*n* = 4). (**K**) ROS levels were analyzed in podocytes transfected with *HIF-1α* siRNA and then incubated with or without fructose (5 mM), polydatin (80 μM), and allopurinol (100 μM) by labeling fluorogenic probe DCFH_2_-DA, evaluated by microplate reader (*n* = 8–10). Data are expressed as the mean ± S.E.M. ^+^
*p* < 0.05, ^++^
*p* < 0.01, ^+++^
*p* < 0.001 compared with normal control group or negative control group, * *p* < 0.05, ** *p* < 0.01, *** *p* < 0.001 compared with fructose vehicle group, ^#^
*p* < 0.05, ^##^
*p* < 0.01 compared with fructose vehicle and *HIF-1α* siRNA transfection group.

**Figure 3 pharmaceutics-14-02202-f003:**
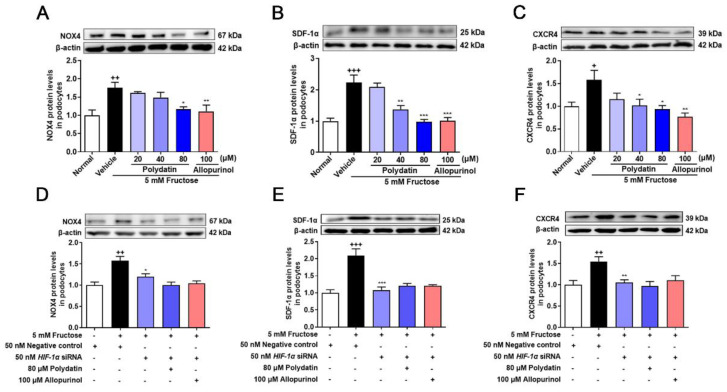
Polydatin suppresses HIF-1α/NOX4 pathway activation in fructose-cultured podocytes. (**A**–**C**) Relative protein levels of NOX4 (24 h) (*n* = 4–5) (**A**), SDF-1α (72 h) (*n* = 6–7) (**B**), and CXCR4 (72 h) (*n* = 4–5) (**C**) were detected in podocytes cultured with or without fructose (5 mM) in the presence or absence of polydatin (20, 40, and 80 μM) or allopurinol (100 μM) by Western blot and quantified by Image J. (**D**–**F**) Relative protein levels of NOX4 (48 h) (*n* = 6) (**D**), SDF-1α (48 h) (*n* = 5–6) (**E**), and CXCR4 (48 h) (*n* = 5–6) (**F**) were detected in podocytes transfected with *HIF-1α* siRNA and then incubated with or without fructose (5 mM), polydatin (80 μM), and allopurinol (100 μM) by Western blot and quantified by Image J. Data are expressed as the mean ± S.E.M. ^+^
*p* < 0.05, ^++^
*p* < 0.01, ^+++^
*p* < 0.001 compared with normal control group or negative control group, * *p* < 0.05, ** *p* < 0.01, *** *p* < 0.001 compared with fructose vehicle group.

**Figure 4 pharmaceutics-14-02202-f004:**
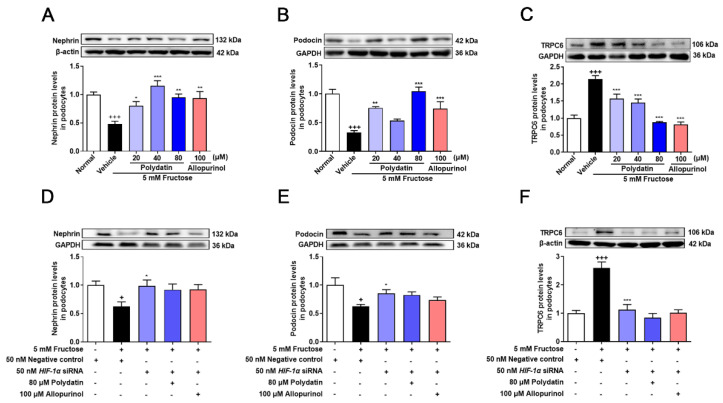
Polydatin prevents high fructose-induced podocyte structural damage. (**A**–**C**) Relative protein levels of nephrin (72 h) (*n* = 6) (**A**), podocin (72 h) (*n* = 5–7) (**B**), and TRPC6 (72 h) (*n* = 6) (**C**) were detected in podocytes cultured with or without fructose (5 mM) in the presence or absence of polydatin (20, 40, and 80 μM) or allopurinol (100 μM) by Western blot and quantified by Image J. (**D**–**F**) Relative protein levels of nephrin (48 h) (*n* = 4–5) (**D**), podocin (48 h) (*n* = 5) (**E**), and TRPC6 (48 h) (*n* = 5–8) (**F**) were detected in podocytes transfected with *HIF-1α* siRNA and then incubated with or without fructose (5 mM), polydatin (80 μM) and allopurinol (100 μM) by Western blot and quantified by Image J. Data are expressed as the mean ± S.E.M. ^+^
*p* < 0.05, ^+++^
*p* < 0.001 compared with normal control group or negative control group, * *p* < 0.05, ** *p* < 0.01, *** *p* < 0.001 compared with fructose vehicle group.

**Figure 5 pharmaceutics-14-02202-f005:**
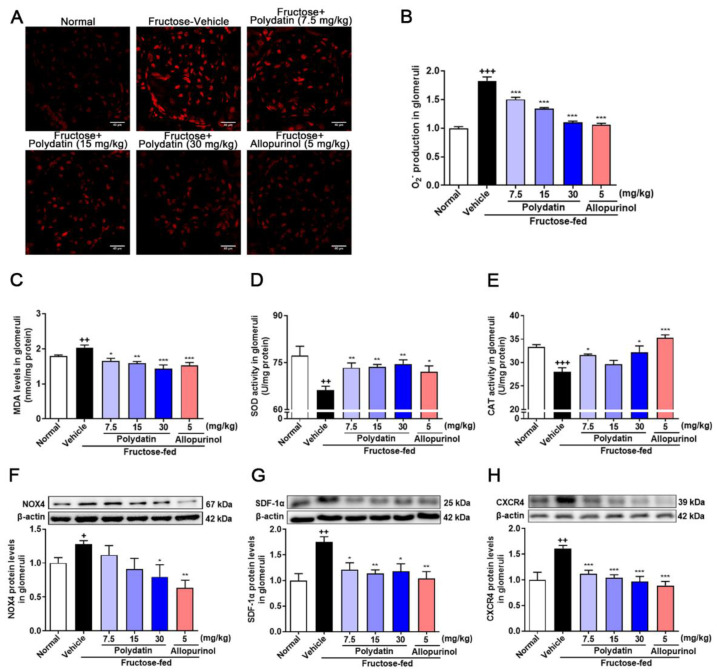
Polydatin prevents oxidative stress of glomeruli in high fructose-fed rats. Rats were fed 10% fructose water (wt/vol) for 13 weeks and treated with polydatin (7.5, 15, and 30 mg/kg) and allopurinol (5 mg/kg) for 7 weeks. (**A**,**B**) O_2_^−^ levels were measured in rat glomeruli by fluorogenic probe DHE (red, scale bar = 40 μm) (*n* = 3) and photographed by a confocal laser scanning microscope. (**C**–**E**) MDA levels (*n* = 5–8) (**C**), SOD activity (*n* = 6–7) (**D**) and CAT activity (*n* = 5–7) (**E**) were measured in rat glomeruli respectively. (**F**–**H**) Relative protein levels of NOX4 (*n* = 4–6) (**F**), SDF-1α (*n* = 4–5) (**G**) and CXCR4 (*n* = 4–5) (**H**) were detected in rat kidney glomeruli by Western blot and quantified by Image J. Data are expressed as the mean ± S.E.M. ^+^
*p* < 0.05, ^++^
*p* < 0.01, ^+++^
*p* < 0.001 compared with normal control group, * *p* < 0.05, ** *p* < 0.01, *** *p* < 0.001 compared with fructose vehicle group.

**Figure 6 pharmaceutics-14-02202-f006:**
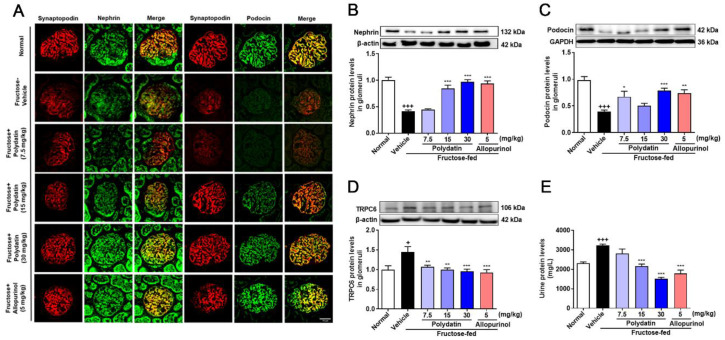
Polydatin attenuates glomerular podocyte structure damage in high fructose-fed rats. Rats were fed 10% fructose water (wt/vol) for 13 weeks and treated with polydatin (7.5, 15, and 30 mg/kg) and allopurinol (5 mg/kg) for 7 weeks. (**A**) Representative immunofluorescence images of kidney glomeruli stained with nephrin (green) and podocin (green) and synaptopodin (red) were examined by a confocal laser scanning microscope (scale bar, 25 μm) (*n* = 3). (**B**–**D**) Relative protein levels of nephrin (*n* = 5–8) (**B**), podocin (*n* = 6–7) (**C**), and TRPC6 (*n* = 4–5) (**D**) were detected in rat glomeruli by Western blot and quantified by Image J. (**E**) Rat urine protein levels were detected by biochemical kit (*n* = 6–7). Data are expressed as the mean ± S.E.M. ^+^
*p* < 0.05, ^+++^
*p* < 0.001 compared with normal control group, * *p* < 0.05, ** *p* < 0.01, *** *p* < 0.001 compared with fructose vehicle group.

**Figure 7 pharmaceutics-14-02202-f007:**
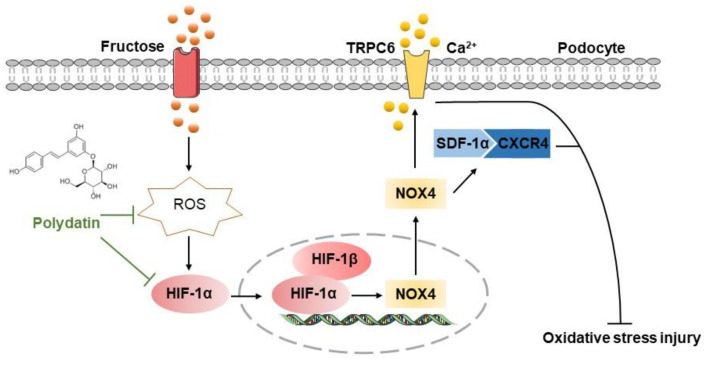
The hypothetical mechanism by which polydatin ameliorates high fructose-induced glomerular podocyte oxidative stress damage. Polydatin protected against high fructose-induced glomerular podocyte oxidative stress. It decreased ROS to down-regulate HIF-1α and NOX4 and then inhibited the activation of the SDF-1α/CXCR4 axis to up-regulate nephrin and podocin and down-regulate TRPC6, resulting in the improvement of podocyte injury.

**Table 1 pharmaceutics-14-02202-t001:** Primer and siRNA sequences.

ID	Sense Primer (5’→3’)	Antisense Primer (5’→3’)
HIF-1α (human)	ATCCATGTGACCATGAGGAAATG	TCGGCTAGTTAGGGTACACTTC
β-actin (human)	CTACCTCATGAAGATCCTCACCGA	TTCTCCTTAATGTCACGCACGATT
HIF-1α (rat)	CCTACTATGTCGCTTTCTTGG	TGTATGGGAGCATTAACTTCAC
β-actin (rat)	GAGAGGGAAATCGTGCGT	GGAGGAAGAGGATGCGG
HIF-1α siRNA	GCGAAGUAAAGAAUCUGAA	UUCAGAUUCUUUACUUCGC

**Table 2 pharmaceutics-14-02202-t002:** Polydatin ameliorates kidney dysfunction in high fructose-fed rats.

Group	Dose (mg/kg)	Serum			Urine
		Uric Acid (μmol/L)	Creatinine (μmol/L)	Urea Nitrogen (mmol/L)	Creatinine (mmol/L)
Normal control	-	94.13 ± 3.90	81.08 ± 4.47	3.06 ± 0.05	19.68 ± 1.79
Fructose vehicle	-	162.25 ± 9.29 ^+++^	136.53 ± 7.72 ^+++^	4.47 ± 0.18 ^+++^	4.96 ± 0.40 ^+++^
Polydatin	7.5	157.63 ± 10.79	125.44 ± 9.10	3.98 ± 0.28	11.90 ± 0.59 ***
Polydatin	15	149.63 ± 9.95	113.59 ± 6.56	2.89 ± 0.15 ***	16.12 ± 1.07 ***
Polydatin	30	116.25 ± 6.15 **	98.81 ± 7.33 **	2.69 ± 0.13 ***	16.51 ± 0.99 ***
Allopurinol	5	117.00 ± 6.79 **	89.96 ± 3.24 ***	2.55 ± 0.17 ***	13.08 ± 0.43 ***

The levels of serum uric acid, serum creatinine, serum urea nitrogen, and urine creatinine were detected in rats (*n* = 5–8), respectively. Data are expressed as the mean ± S.E.M. ^+++^
*p* < 0.001 compared with normal control group, ** *p* < 0.005, *** *p* < 0.001 compared with fructose-vehicle group.

## Data Availability

The data presented in this study are available upon request from the corresponding author.

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
