# Peer review of "Polydatin Ameliorates High Fructose-Induced Podocyte Oxidative Stress via Suppressing HIF-1α/NOX4 Pathway"

_pharmaceutics, 2022, doi:10.3390/pharmaceutics14102202_

Round 1

Reviewer 1 Report

After carefully review of the manuscript, I have found that all the experiments are conducted appropriately and it has been written nicely. However, I have minor concerns that should be addressed before the acceptance of the manuscript in ‘Pharmaceutics’.

1.      The authors have mentioned as, ‘Doses of polydatin and allopurinol were selected based on our previous studies and other reports [5,27-29].’ However, no study cited in the text (5,27-29) has reported the use of ‘Allopurinol’. Therefore, the authors are suggested to cite the appropriate references related to the doses of Allopurinol. Besides, the authors are also suggested to justify these cited references or avoid self-citing some that are needed for the submitted manuscript.

2.      Please provide the original blot of 1E in the supplementary materials.

Author Response

Reply for comment 1: 

Thanks for your constructive comments. References to doses of polydatin and allopurinol may be ambiguous. The reference of 27 (now 28) has reported the dose of allopurinol in fructose-fed rats. We readjusted and cited the appropriate references related to the doses of allopurinol. We carefully checked the necessity and accuracy of reference citations in the submitted manuscript to avoid self-citing.

Reply for comment 2: 

As suggested, we provided the original blot of 1E in the supplementary materials.

Reviewer 2 Report

Manuscript “Polydatin ameliorates high fructose-induced podocyte oxidative stress via suppressing HIF-1α/NOX4 pathway” represents a contribution to field of pharmaceutics. Text is clear and easy to read. Conclusions are consistent with the evidence and arguments presented. The research topic is original.

Before accepting the manuscript, it is essential that the authors:

-                  Separate the conclusion as a separate paragraph.

-          It is known in the literature that fructose consumption reduces hepatocyte NADPH oxidase 4 (NOX4), please cite the literature: Bettaieb, A.; Jiang, J.X.; Sasaki, Y.; Chao, T.I.; Kiss, Z.; Chen, X.; Tian, J.; Katsuyama, M.; Yabe-Nishimura, C.; Xi, Y.; et al. Hepatocyte nicotinamide adenine dinucleotide phosphate reduced oxidase 4 regulates stress signaling, fibrosis, and insulin sensitivity during development of steatohepatitis in mice. Gastroenterology 2015, 149, 468–480.

-             Line 582, ´These results propose a new idea that suppression of HIF-1α/NOX4 pathway by polydatin may be a novel strategy for oxidative stress-related podocyte injury.´ The use of polydatin in oxidative stress is known in the literature: https://doi.org/10.1155/2015/362158, https://doi.org/10.1155/2022/9218738, etc. Please reformulate your conclusion.

Author Response

Reply for comment 1:

As suggested, we have separated the conclusion as a separate paragraph in the manuscript.

Reply for comment 2:

Thanks for your significant reminding. The study you mentioned explored that the fast food diet (supplemented with high-fructose corn syrup) induced NOX4 high expression, causing oxidative stress, lipid peroxidation, inflammation, and apoptosis in mouse hepatocytes. The upregulation of NOX4 in high fructose-exposed podocytes was also observed in our study. So, we add the literature in line 60.

Reply for comment 3:

Thanks for your valuable suggestions. We carefully read the references you mentioned and then cited in line 539. We rigorously reformulate the part of conclusion in the manuscript to better express our thoughts.